

# Icariin activates far upstream element binding protein 1 to regulate hypoxia-inducible factor-1α and hypoxia-inducible factor-2α signaling and benefits chondrocytes

Pengzhen Wang[1,*], Pingping Zhu[2,*], Shaoheng Zhang[3], Wei Yuan[4] and Zhihe Liu[1]

[1] Guangzhou Institute of Traumatic Surgery, Guangzhou Red Cross Hospital of Jinan University, Guangzhou, China
[2] Department of Neurology, Guangzhou Red Cross Hospital of Jinan University, Guangzhou, Guangdong, China
[3] Department of Cardiology, Guangzhou Red Cross Hospital of Jinan University, Guangzhou, China
[4] Department of Hepatobiliary Surgery, Guangzhou Red Cross Hospital of Jinan University, Guangzhou, Guangdong
* These authors contributed equally to this work.

Corresponding authors
Pengzhen Wang,
wang521jnu@163.com
Zhihe Liu, zliu0731@163.com

## ABSTRACT

Icariin (ICA) is a typical flavonoid glycoside derived from epimedium plants. It has both anabolic and anti-catabolic effects to improve bone mineral density and reduce bone microstructural degradation. However, the effect and underlying mechanism of ICA on the proliferation and metabolism of chondrocyte and synthesis of extracellular matrix are still unclear. This study aimed to investigate the role and regulation of far upstream element binding protein 1 (FUBP1) in chondrocytes treated with ICA to maintain homeostasis and suppress inflammatory responses. In the study, the effect of ICA on chondrocytes with overexpressed or silenced FUBP1 was detected by the MTS and single-cell cloning methods. The expression of hypoxia-inducible factor-1/2α (HIF-1/2α), FUBP1, matrix metalloproteinase (MMP)9, SRY-box transcription factor 9 (SOX9), and type II collagen (Col2α) in ATDC5 cells, a mouse chondrogenic cell line, treated with ICA was evaluated by immunoblotting. Western blotting revealed 1 μM ICA to have the most significant effect on chondrocytes. Alcian blue staining and colony formation assays showed that the promoting effect of ICA was insignificant in FUBP1-knockdown cells ($P > 0.05$) but significantly enhanced in FUBP1-overexpressed cells ($P < 0.05$). Western blot results from FUBP1-knockdown cells treated with or without ICA showed no significant difference in the expression of FUBP1, HIF-1/2α, MMP9, SOX9, and Col2α proteins, whereas the same proteins showed increased expression in FUBP1-overexpressed chondrocytes; moreover, HIF-2α and MMP9 expression was significantly inhibited in FUBP1-knockdown chondrocytes ($P < 0.05$). In conclusion, as a bioactive monomer of traditional Chinese medicine, ICA is beneficial to chondrocytes.

## INTRODUCTION

Articular cartilage injury is a frequently-occurring disease that is more common in the knees and hips (*Borrelli et al., 2019*). Articular cartilage injury commonly causes joint swelling, pain, and deformity because of the poor self-healing ability of cartilage (*Tanaka, 2020*). The development and progress of early arthritis exert enormous economic and social pressure on patients, their families, and society. Therefore, it is important to find efficient and cheap drugs to treat articular cartilage injury and to speed up the research on its pathogenesis.

When articular cartilage is subjected to pressure wear or mechanical damage, reduced chondrocyte activity is accompanied by inflammation and apoptosis in chondrocytes, reduced expression of the extracellular matrix (ECM) regulator SRY-box transcription factor 9 (SOX9), and increased expression of matrix metalloproteinase 9 (MMP9) in chondrocytes (*Rong et al., 2019*). Currently, both surgical and non-surgical treatments are administered for cartilage damage. Surgical treatment of cartilage injury includes drilling microfractures, chondrocyte transplantation, and joint replacement (*Ferreira et al., 2019*), although surgical treatment is considered to be prone to recurrence and is associated with high cost and risk. Non-surgical treatment involves the administration of non-steroidal anti-inflammatory analgesics that can only delay disease progression but cannot fundamentally cure osteoarthritis (*Lin et al., 2016*). Different from steroidal and non-steroidal drugs, Chinese medicinal monomers can inhibit chondrocyte apoptosis, cellular ECM degradation, and inflammatory factor expression, and promote chondrocyte proliferation through intracellular signaling pathways such as Akt/mammalian target of rapamycin (mTOR) (*Tang et al., 2021*) and nuclear factor kappa-light-chain-enhancer of activated B cells (NF-κB) pathways (*Mi et al., 2018*) to exert a comprehensive effect on cartilage.

Previously, a scientific report suggested that natural Chinese medicine therapies could attenuate bone resorption and promote bone formation for osteoporosis treatment (*Wang et al., 2017*). ICA is a typical flavonoid glycoside with an isopentenyl group connected to C-8 on the molecule and is characterized by medicinal and edible properties (*Mi et al., 2018*). ICA is mainly used for the treatment of osteoporosis in Asian countries such as China and Japan because of its low price and remarkable efficacy. There are few studies on the protective effect and mechanism of ICA on chondrocytes. As an important transcription factor in chondrocytes, HIF-1α is a regulatory gene for chondrocyte activity and ECM secretion (*Hu et al., 2020*). Maintaining a high level of HIF-1α expression in chondrocytes is an important molecular means for the treatment and repair of cartilage damage. HIF-2α, homologous to HIF-1α, reportedly also plays a key regulatory role in the metabolism of articular chondrocytes. A recent study suggested that HIF-2α is a key factor in maintaining cartilage homeostasis (*Cheng et al., 2015*). Interleukin (IL)-1β, NF-κB (p65), c-Jun N-terminal kinase (JNK), SOX9, MMP13, and a disintegrin and

metalloproteinase with thrombospondin motifs 5 (ADAMTS5) are common molecules that promote ECM decomposition and synthesis of regulatory genes (*Fan et al., 2022*). Our previous research confirmed that ICA can promote the accumulation of HIF-1α, inhibit the expression of HIF-2α in chondrocytes to maintain chondrocyte viability, and promote cartilage ECM synthesis (*Wang et al., 2016*, *2020b*, *2020c*).

Far upstream element binding protein 1 (FUBP1), containing 644 amino acids and having helicase activity, is responsible for regulating cell proliferation by targeting c-Myc (*Ma et al., 2020*). FUBP1 is known to participate in the malignant process and glycolysis of colon cancer cells by combining with c-Myc (*Wang et al., 2022*). It promotes the proliferation of lung squamous cell carcinoma cells and regulates tumor immunity through programmed death-ligand 1 (PD-L1) (*Yu et al., 2022*). Previously, we confirmed ICA-mediated inhibition of FUBP1 expression in SKOV3, a human ovarian cancer cell line (*Wang et al., 2019*) Thus far, FUBP1 expression and function in chondrocytes have not been fully studied, and the relationship between FUBP1 and HIF-1α/2α in chondrocytes is unclear. In this study, we aimed to clarify the relationship between FUBP1 and HIF-1α/2α in chondrocytes. This research will establish a solid foundation for future research.

# MATERIALS AND METHODS

## Reagents

The main chemical component in ICA is $C_{33}H_{40}O_{15}$ (Fig. 1A), which was purchased from Beijing Bio Rule Biotechnology Co., Ltd (#489-32-7; Beijing, China). ICA was frozen as a 0.1 M stock solution at −20 °C.

## MTS

ATDC5 cells, a mouse chondrogenic cell line, were donated by Professor Yang Xuesong from the Medicine College of Jinan University. The cells were divided into four groups according to ICA treatment: control (Con), 0.5, 1, and 2 μM ICA. Cells were seeded at 3,000 cells per well in a 5% $CO_2$ incubator for 24 h. These chondrocytes were mixed with 20 μL of MTS reagent (#G1112; Promega, Madison, WI, USA) followed by culture for 4 h before analyzing using the GloMax Multi+ Detection System (Promega, Madison, WI, USA) by measuring their absorbance values at 490 nm.

## Representative brightfield cell images

Four groups of cells according to ICA treatment: Con, 0.5, 1, and 2 μM ICA were seeded at $\times 10^5$ cells per well in a 5% $CO_2$ incubator for 24 h. Brightfield photographs of the cells were obtained using an inverted microscope (Olympus, Shinjuku City, Tokyo, Japan).

## FUBP1 knockdown and overexpression in chondrocytes

A pSi-LVRU6GP vector with a puromycin resistance cassette (GeneCopoeia, Rockville, MD, USA) was used to express short hairpin (sh) RNA to knock down FUBP1 expression. FUBP1 overexpression and FUBP1 deficiency were induced by transfecting the cells with vectors encoding FUBP1 and shRNA, respectively; control cells were transfected with

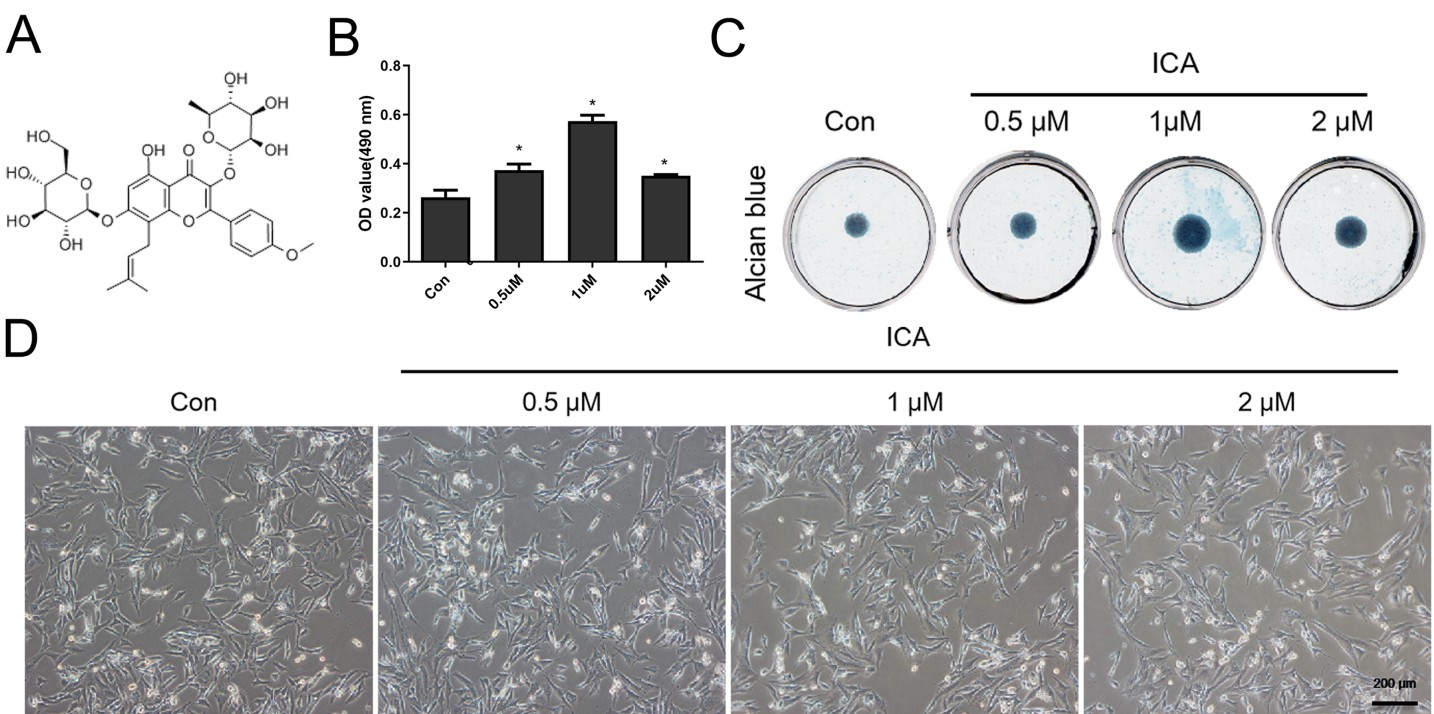

**Figure 1 Screening for the optimal concentration of ICA for chondrocyte treatment.** (A) Chemical structural formula of ICA. (B) MTS assay of chondrocytes treated with or without ICA (0, 0.5, 1, and 2 μM) for 1 day. (C) Alcian blue staining of cell cultures for 14 days. (D) The lack of morphological differences between the control and ICA-treated cell groups. Magnification 100×. *$P < 0.05$ vs. control group, $n = 3$.

control vectors. The cell lines stably expressing FUBP1 or with stably knocked down FUBP1 were named FUBP1-OE and FUBP1-KD cell lines, respectively.

## Micro-mass culture and Alcian blue staining

The cells were divided into six groups: Con, ICA-treated, FUBP1-KD, ICA-treated FUBP1-KD, FUBP1-OE, and ICA-treated FUBP1-OE. For the experiment, the Con cells were exposed to 1× phosphate-buffered saline (PBS) and FUBP1-KD and FUBP1-OE cells were exposed to 1 μM ICA for 24 h. The experimental procedure of micro-mass culture has been described previously (*Wang et al., 2016*). The scanner acquired photomicrographs of stained cell clusters. The ECM aggrecan (ACAN) stained blue.

## Colony formation assay

In total, 500 cells from six cell groups (Con, ICA-treated, FUBP1-KD, ICA-treated FUBP1-KD, FUBP1-OE, and ICA-treated FUBP1-OE) were transferred to six well-plates. The experimental procedure for crystal violet staining has been described previously (*Wang et al., 2016*). Cell clone photos were obtained with a scanner. A cell mass containing more than 50 cells is called a cell colony.

## Cell and scaffold three-dimensional complex construction and culture

The sterile gelatin ($3 \times 3 \times 3$ mm$^3$) was UV-irradiated for 2 h on an ultra-clean bench. All six sides of the gelatin (#190608; Nanjing Pharmaceutical, Nanjing, China) were

poured into 30 μL of the alginate solution (#9005-38-3; Sigma, St. Louis, MO, USA) containing $3 \times 10^6$ cells/mL. After 3 h, the cells were adhered to form complex blocks and were cultured for 14 days.

For the three-dimensional complexes, the complexes were washed twice with ice-cold PBS, fixed with 4% paraformaldehyde for 24 h, dehydrated with a sucrose gradient solution, and embedded within the optimal cutting temperature compound (#4583; Sakura, Torrance, CA, USA). The complexes were frozen in liquid nitrogen and 10 μm thin frozen sections were obtained using the cryostat and stored at 80 °C. The sections were harvested for subsequent safranin O (SO)/fast green and hematoxylin and eosin (H&E) staining (#G1371; Solarbio, Beijing, China). Briefly, for H&E staining, the sections were immersed in hematoxylin dye for 10 min and rinsed with tap water for 2 min. After rinsing with 1% alcohol hydrochloride for 30 s and tap water for 2 min, the sections were infiltrated by the eosin solution for 5 min and rinsed with tap water for 2 min. Finally, the slices were sealed with neutral gum. For SO staining, the nuclei were stained with hematoxylin solution, then the sections were immersed in 1% SO for 10 min. After exposing to 1% glacial acetic acid for 15 s, the sections were stained with 0.1% SO for 10 min and fixed with neutral gum.

## Western blotting

The cells were divided into six treatment groups: Con, ICA-treated, FUBP1-KD, ICA-treated FUBP1-KD, FUBP1-OE, and ICA-treated FUBP1-OE. During the experiment, the cells at a concentration of $2.0 \times 10^5$ cells/well were exposed to $1 \times$ PBS (Con, FUBP1-KD, and FUBP1-OE) or 1 μM ICA (ICA-treated FUBP1-KD and ICA-treated FUBP1-OE) for 24 h in 6-well plates, respectively. Protein concentrations were analyzed using the BCA kit (#P0009; Beyotime, Shanghai, China) and 30 μg of protein per well was electrophoresed on a 10% sodium dodecyl sulfate-polyacrylamide gel. Thereafter, the proteins were transferred to a polyvinylidene difluoride membrane. After blocking the unbound sites with a non-specific protein, the membranes were incubated overnight at 4 °C with a primary rabbit antibody against glyceraldehyde 3-phosphate dehydrogenase (GAPDH; 1:1,000; #5174; CST, Danvers, MA, USA), HIF-1α (1:1,000; #36169; CST, Danvers, MA, USA), FUBP1 (1:200; #ab181111; Abcam, Cambridge, United Kingdom), HIF-2α (1:1,000; #71565; Cell Signaling Technology, Danvers, MA, USA), MMP9 (1:1,000; #13667, CST, Danvers, MA, USA), Col2α (1:1,000; #ab34712), and SOX9 (1:1,000; #82630, CST). Next, the membranes were incubated with horseradish peroxidase-labeled secondary antibody goat anti-rabbit IgG (1:3,000; #ARG 65351; Arigo Biolaboratories, Burlington, Ontario, Canada) for 1 h next day. The protein bands were detected with ECL-Plus detection systems (#1705060, Bio-Rad, Hercules, CA, USA) and quantified using Image Lab system version 2.0 (Bio-Rad, Hercules, CA, USA).

## Immunofluorescence staining

The cells were divided into three groups: FUBP1-Con, FUBP1-KD and FUBP1-OE. The slides were treated with poly-L-lysine and then placed in a 6-well plate. The chondrocytes at a concentration of $2.0 \times 10^4$ cells/well were seeded in 6-well plates for 24 h.

After 24 h, the medium was removed. Chondrocytes were fixed with 4% paraformaldehyde for 20 min, then cells were immersed with 0.5% Triton X-100 for 10 min for perforating cell membranes and washed three times with PBS, and then, the slides were blocked with serum for 30 min, a rabbit anti-FUBP1 (1:200) antibody solution was added on the slides in the 6-well plate for overnight incubation at 4 °C. After washing, the cells were incubated with HRP-labeled secondary antibodies for 1 h at room temperature. The target protein was visualized using a fluorescence microscope (CLSM; Zeiss LSM 510 META System).

## Quantitative polymerase chain reaction

The chondrocytes were divided into three groups: FUBP1-Con, FUBP1-KD and FUBP1-OE. The chondrocytes at a concentration of $2.0 \times 10^5$ cells/well were seeded in 6-well plates for 24 h. After 24 h, the medium was removed. Total RNA was extracted from chondrocytes using TRIzol reagent (Thermo Fisher Scientific, Waltham, MA, USA) and converted to cDNA using a kit named PrimeScript RT Master Mix (Takara Bio, Kusatsu, Shiga, Japan). qPCR was performed with SYBR Premix ExTaq (Takara Bio, Kusatsu, Shiga, Japan) in a qTOWER 3G Real-time PCR system (Analytik Jena, Jena, Germany). Primer sequences used were as follows: FUBP1 forward, 5′-GGAACAACACCTGATAGGATAGC-3′ and reverse, 5′-GCCAGCCTGAACACTTCGTAG-3′; GAPDH forward, 5′-AGGTCGGTGTGAACGGATTTG-3′ and reverse, 5′-TGTAGACCATGTAGTTGAGGTCA-3. The relative gene expression levels were quantified using the $2^{-\Delta\Delta CT}$ method and normalized to the internal reference gene GAPDH.

## Statistical analysis

All samples were performed in triplicate. Data (mean ± standard deviation) were analyzed by using a t-test for two groups and one-way ANOVA for multiple groups using SPSS 22.0 software.

# RESULTS

## Effects of ICA on chondrocyte viability, ECM secretion, and cell morphology

As Fig. 1B reflected, ICA at doses of 0.5, 1 and 2 μM significantly increased the chondrocyte viability, as compared with control cells ($P < 0.05$), moreover, the effect of 1uM ICA on chondrocytes was the most obvious. The result (Fig. 1C) indicated that 0.5, 1 and 2 μM of ICA increased the ACAN secretion of chondrocytes at varying degrees. However, 1 μM ICA has the most significant effect on chondrocytes to promote ACAN synthesis, which was consistent with Fig. 1B. The result (Fig. 1D) indicated that the cell morphology was not changed by ICA. Therefore, 1 μM ICA was recommended for subsequent research.

## Role of FUBP1 in ICA-treated chondrocyte proliferation and ECM synthesis

Figure 2A showed that chondrocytes with FUBP1 (green) overexpression (FUBP1-OE) and knockdown (FUBP1-KD) were successfully constructed. The fluorescence intensity of

FUBP1 (green) in FUBP1-OE and FUBP1-KD groups was stronger and weaker than that of the control groups, respectively. Figure 2B showed that FUBP1 was overexpressed and knockdown in chondrocytes successfully. Figure 2C indicated ICA significantly promoted ECM synthesis. However, the promotion effect of ICA was not significant in FUBP1 knockdown cells, and the promotion effect of ICA was significantly enhanced in FUBP1 overexpressed cells. As Figs. 2D and 2E showed, the ICA groups contained a higher number of cell clones compared to the control group. For cell cloning experiments, the promotion effect of ICA was significantly enhanced in FUBP1 overexpressed cells ($P < 0.05$).

### Role of FUBP1 in chondrocytes cultured in the alginate gelatin scaffold

As Fig. 3 shows, the ICA-treated cell groups exhibited a higher ACAN synthesis by chondrocytes than the control groups ($P < 0.05$). The difference in ACAN synthesis in FUBP1-KD cells including control groups and ICA-treated cell groups was statistically insignificant.

### Role of FUBP1 in HIF-1α and HIF-2α signaling in chondrocytes

As Figs. 4A–4D shows, compared with the control group, the protein expression of FUBP1, HIF-1α, SOX9, and Col2α was increased significantly in the ICA-treated cell group ($P < 0.05$) as well as ICA-treated FUBP1-OE cell group ($P < 0.05$). the expression of these molecules in FUBP1-knockdown cells was unaffected. Conversely, as Figs. 4E–4H reveals, the western blotting and quantitative analysis showed that HIF-2α and MMP9 protein expression was reduced significantly in the ICA-treated groups than in the control group ($P < 0.05$). The difference in HIF-2α and MMP9 expression in FUBP1-KD cells treated with or without ICA was insignificant. Thus, overexpression of FUBP1 in chondrocytes significantly inhibited HIF-2α and MMP9 expression in the ICA-treated groups ($P < 0.05$).

## DISCUSSION

This study revealed that ICA promotes chondrocyte ECM synthesis and chondrocyte proliferation by targeting FUBP1 to activate HIF-1α and inactivate HIF-2α. However, after FUBP1 knockdown, the protective effect of ICA on chondrocytes disappeared, whereas, after FUBP1 overexpression, the protective effect of ICA on chondrocytes was significantly enhanced; thus, ICA-mediated pro-proliferative and pro-ECM synthesis effects on chondrocytes are mediated by FUBP1. This study clarified the mechanism of ICA on chondrocyte proliferation and ECM synthesis and will provide a theoretical basis for the future use of ICA the clinical treatment of cartilage-related diseases.

In recent years, traditional Chinese medicine has become increasingly popular in disease treatment (*Shen et al., 2022*; *Wang et al., 2020a*, *2020d*). As an active flavonoid glycoside (*El-Shitany & Eid, 2019*), ICA has a wide range of pharmacological effects, including anti-myocardial infarction (*Zeng et al., 2022*), anti-neuronal aging (*Jin et al., 2019*), prevention and treatment of osteoporosis (*Huang et al., 2020*; *He et al., 2018*), and inhibition of inflammation (*Su et al., 2018*) and anticancer (*Gu et al., 2017*). In recent years,

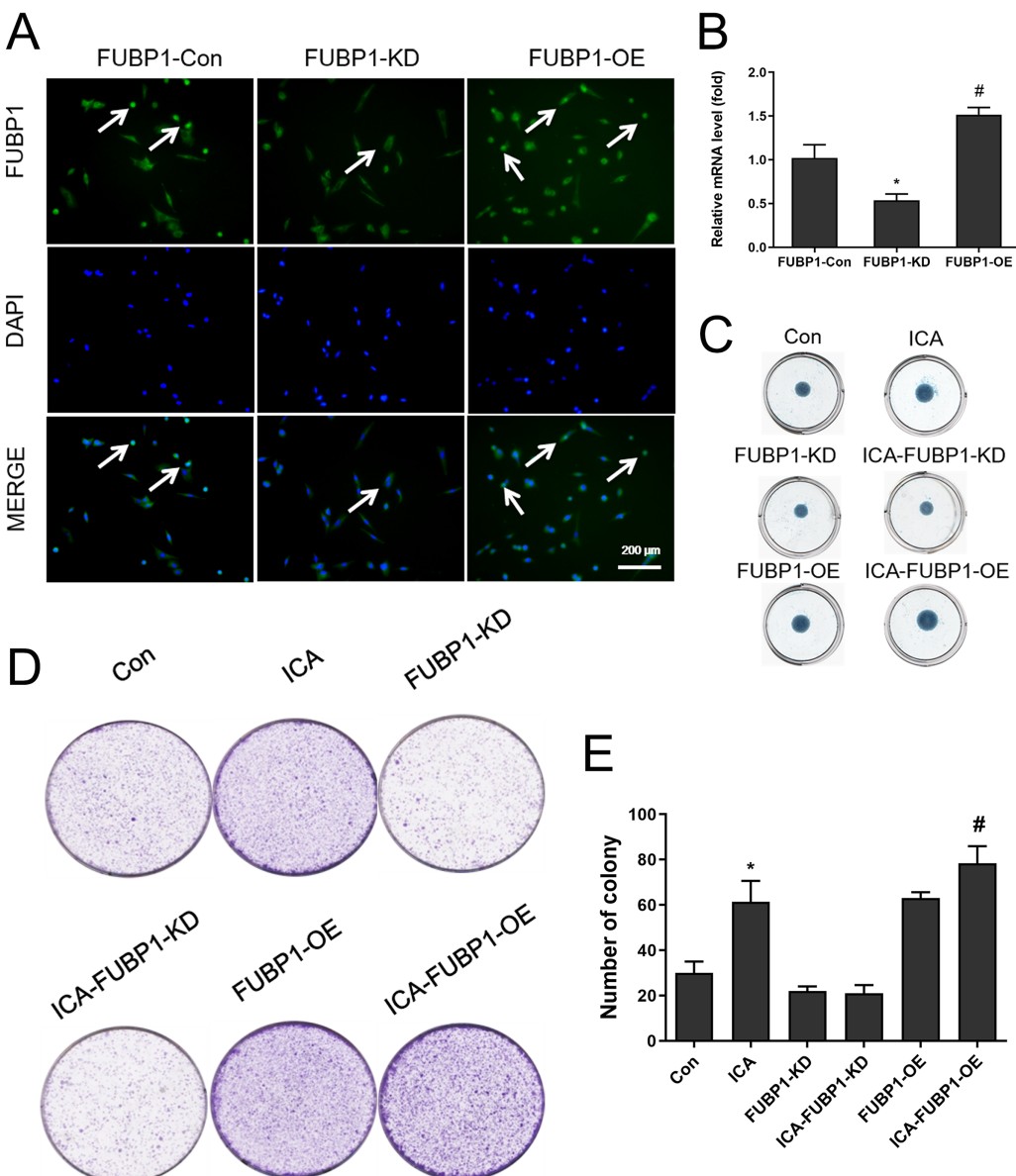

**Figure 2 The effect of FUBP1 on chondrocyte proliferation and extramatrix synthesis.** (A) FUBP1 expression in chondrocytes from control, FUBP1 overexpression (FUBP1-OE) and FUBP1 knockdown (FUBP1-KD) groups determined by immunofluorescence with anti-FUBP1 (green). DAPI was stained in blue. Magnification 100×. (B) The mRNA expression of FUBP1 in chondrocytes from control, FUBP1 overexpression (FUBP1-OE) and FUBP1 knockdown (FUBP1-KD) groups determined by q-PCR method. (C) Chondrocytes in six groups (Con, ICA, FUBP1-KD, ICA-FUBP1-KD, FUBP1-OE and ICA-FUBP1-OE) were processed for alcian blue staining. (D) The ability of all groups of chondrocytes to form clones was tested by cell clony and crystal violet staining. (E) Quantitative analysis of the number of cell clones in (C). In (B), $^*P < 0.05$ is used to mark the significant difference between the FUBP1-Con group and the FUBP1-KD group. $^\#P < 0.05$ is used to mark the significant difference between the FUBP1-Con group and the FUBP1-OE group. In (E), $^*P < 0.05$ is used to mark the significant difference between the control group and the ICA group. $^\#P < 0.05$ is used to mark the significant difference between the control group and the ICA group in FUBP1-OE chondrocytes, $n = 3$.

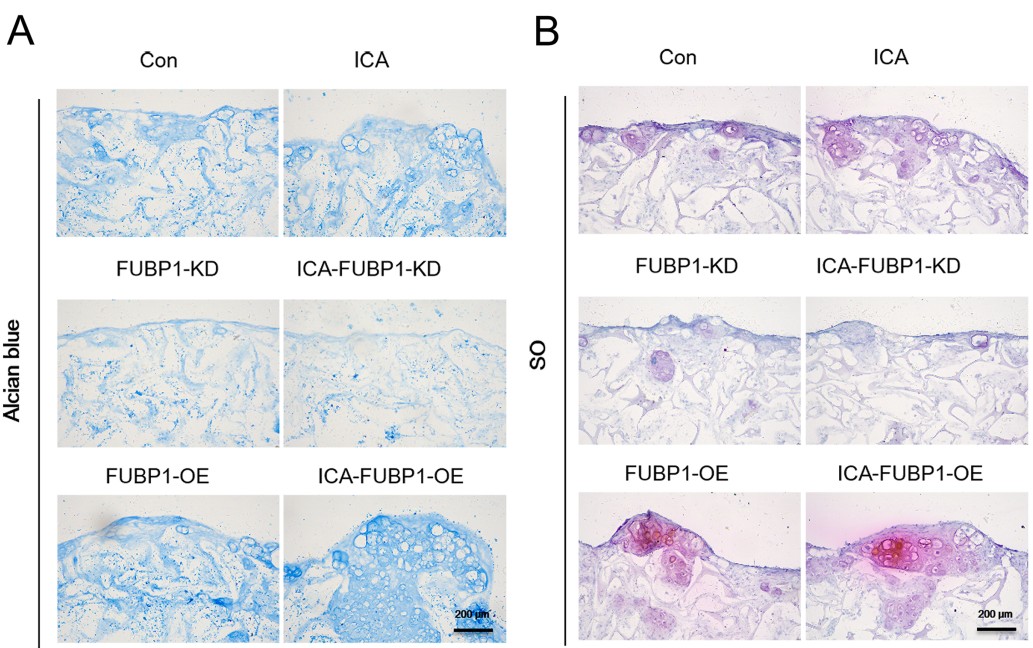

**Figure 3 FUBP1 potentiates cartilage regeneration from chondrocytes stimulated by ICA.** Alcian blue (A) and SO (B) stained histological images of chondrocytes cultured in the alginate gelatin scaffold three-dimensional culture system treated with ICA (1 μM) or without ICA (control) for 14 days, $n = 3$.

ICA combined with chitosan was found to promote the repair of cartilage defects in rabbits (*Zhang et al., 2020*). The main effect of ICA on chondrocytes is to promote cartilage gene expression and ECM synthesis (*Huang et al., 2019*; *Chen et al., 2022*; *Zhu et al., 2022*). However, the mechanisms underlying these processes have not been elucidated and need to be further explored.

One study showed that ICA has great potential as a candidate growth factor for the treatment of cartilage injury (*Wang et al., 2020e*), it is suggested that ICA protects vertebral endplate chondrocytes against apoptosis and degeneration by activating the nuclear factor erythroid 2–related factor 2 (Nrf-2)/heme oxygenase-1 (HO-1) pathway (*Shao et al., 2022*). We have previously shown that ICA increases glycolytic and chondrocyte activity, and simultaneously inhibits the NF-κB/HIF-2α signaling pathway and inflammatory responses (*Wang et al., 2020b*). In the present study, ICA at 1 μM had the most significant effect on chondrocyte proliferation and ECM synthesis and did not affect chondrocyte morphology, as shown by bright field observation. One study found that the long non-coding RNA (lncRNA) non-imprinted in Prader-Willi/Angelman syndrome region protein 1 (NIPA1)-sense overlapping exerted an inhibitory effect on vascular inflammation by binding to FUBP1 and thereby inhibiting NIPA1 expression (*Jiang et al., 2023*), however, the expression and role of FUBP1 in chondrocytes are still unclear. The current research showed that the beneficial effect of ICA on chondrocytes was nullified after FUBP1 silencing, whereas it was significantly enhanced after FUBP1 overexpression. These results suggest that FUBP1 may be a key regulator of matrix synthesis and homeostatic state maintenance in chondrocytes.

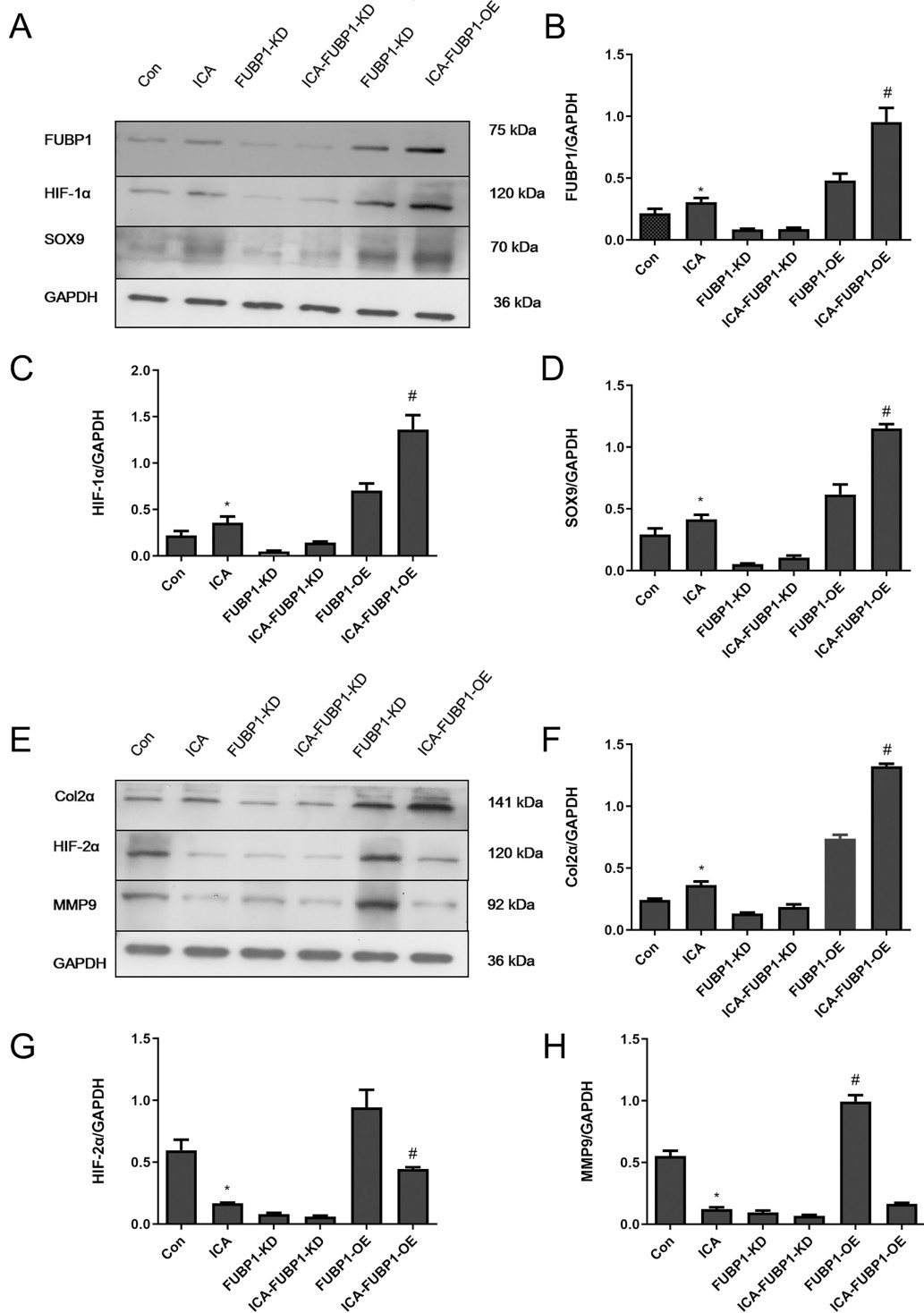

**Figure 4  FUBP1 activates the HIF-1α signaling pathway and inhibits the HIF-2α signaling pathway to benefit chondrocytes.** (A) FUBP1, HIF-1α, and SRY-box transcription factor 9 (SOX9) protein levels from the control (Con), icariin (ICA), FUBP1-knockdown (FUBP1-KD), ICA-FUBP1-KD, FUBP1-overexpressed (FUBP1-OE), and ICA-FUBP1-OE cell groups were evaluated using western blotting. (B–D) Quantitative analysis of the protein bands representing FUBP1, HIF-1α, and SOX9 protein. (E) Col2α, HIF-2α, and MMP9 protein levels from the Con, ICA, FUBP1-KD, ICA-FUBP1-KD, FUBP1-OE, and ICA-FUBP1-OE cell groups were evaluated using western blotting. (F–H) Quantitative analysis of

**Figure 4** (continued)
the protein bands representing Col2α, HIF-2α, and MMP9 protein. $^*P < 0.05$ indicates a significant difference between the control and ICA groups. $^\#P < 0.05$ indicates a significant difference between the control and ICA groups in FUBP1-OE chondrocytes, $n = 3$.

Micro-mass culture was used to study the effects of ICA on chondrocyte proliferation and ECM synthesis in a two-dimensional manner. This study used composite scaffolds composed of gelatin and alginate to simulate the growth microenvironment of chondrocytes *in vivo*. Gelatin and alginate cross-linked scaffolds are known to exhibit a highly flexible structure and good bioactivity, which was a preferred choice for culturing cells *in vitro* to mimic the environment *in vivo* (*Ho et al., 2021*). Alcian blue and SO staining is a classic method to detect ECM secretion from chondrocytes (*Alibegović, Blagus & Martinez, 2020*). The results from the frozen sections of the cells and scaffold complexes showed that FUBP1 silencing abolished the ICA-mediated ECM synthesis-promoting effect on chondrocytes, whereas FUBP1 overexpression enhanced it. The results of the three-dimensional culture are consistent with the results of the monolayer culture of chondrocytes.

HIF-1α, HIF-2α, and HIF-3α are the three existing forms of HIF-α (*Li et al., 2020*). Under abnormal oxygen concentration, prolyl hydroxylase, which specifically degrades HIF, is inactivated; hence, HIF-1α has the opportunity to translocate into the nucleus and form a dimer with HIF-1β, thereby activating related pathway genes (*Chen et al., 2019*). Previous studies have confirmed that flavonoid glycosides such as quercetin and baicalin promote HIF-1α accumulation, and the present study showed that ICA, also a flavonoid glycoside, resulted in increased HIF-1α expression (*Lee & Lee, 2008*; *Wang et al., 2020c*). Col2α, SOX9, and ACAN expression in chondrocytes increased by HIF-1α upregulation (*Li et al., 2018*), whereas in FUBP1-KD cells, the ICA-stimulatory effect on HIF-1α was abolished, and simultaneously, chondrocyte proliferation and ECM secretion were also reduced. Therefore, we infer that the ICA target in chondrocytes is likely to be FUBP1, which warrants further research.

HIF-2α is highly expressed in the osteoarthritic cartilage and is mainly involved in inflammatory responses (*Ito et al., 2021*; *Li et al., 2022*). HIF-2α participates in the progression of cartilage damage *via* NF-κB pathway regulation (*Murahashi et al., 2018*). Cartilage damage is accompanied by chondrocyte apoptosis and senescence (*Roberts et al., 2016*). SOX9 and MMP9 are regulators of ECM metabolism (*Byun et al., 2020*). In addition, the pro-inflammatory cytokines sourced from mechanical injury promote the transition from cartilage damage to osteoarthritis (*Guan et al., 2018*; *Sanchez-Lopez et al., 2022*). Many studies have confirmed that HIF-2α directly regulates the expression of MMP9 (*Wang et al., 2020b*; *Pi et al., 2015*). Our previous study confirmed that ICA inhibited MMP9 expression by inhibiting the expression of HIF-2α expression (*Wang et al., 2020b*). However, after FUBP1 knockdown, MMP9 expression was up-regulated and ECM secretion by chondrocytes was reduced, whereas, after FUBP1 overexpression, the expression patterns were reversed. Therefore, FUBP1 assumes a mediating role in the

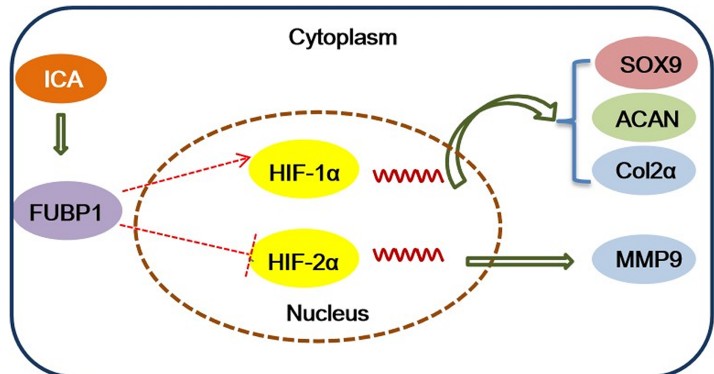

**Figure 5 Schematic model of ICA-induced protective effect on chondrocytes.** ICA promotes the HIF-1α pathway to form cartilage and suppresses the HIF-2α pathway to inhibit extracellular matrix degradation by FUBP1. HIF-1α positively regulates SOX9, Col2α, and ACAN to promote cartilage matrix synthesis, and negatively regulates MMP9 to inhibit cartilage matrix degradation. HIF-1α and HIF-2α cooperate to regulate and maintain a balance in the cartilage matrix.

regulation of HIF-2α in chondrocytes treated with ICA. Although the experimental approach of knockout and overexpression confirmed that ICA could promote HIF-1α expression and inhibit HIF-2α expression by activating FUBP1 in chondrocytes. However, there are some limitations in this study, that is, it is not clear how FUBP1 and HIF-1α interact, and how FUBP1 and HIF-2α interact when ICA stimulates chondrocytes. In the future, we will carry out double luciferase test and immunoprecipitation test to solve this problem.

## CONCLUSIONS

In conclusion, this study identified the important role of FUBP1 in ICA in protecting chondrocytes to maintain a homeostatic state and suppress inflammatory responses. The major mechanistic clues were depicted in Fig. 5. Overexpression or silencing of FUBP1 regulates the HIF-1/2 signaling pathway to maintain chondrocyte activity or prevent chondrocyte inflammatory responses, respectively. In the future, ICA can be used as a candidate drug to protect cartilage.

### Funding

This work was supported by the Medical Science and Technology Research Foundation of Guangdong (A2021335, PW), Traditional Chinese Medicine Bureau of Guangdong Province (20222166, PW), Guangdong Provincial Basic, Applied Basic Regional Joint Fund (2020A1515110009, PZ), Guangzhou Science and Technology Project (202102010060, ZL), Guangzhou Science and Technology Bureau City School (Institute) Enterprise Joint Project (SL2024A03J00901, PW). The funders had no role in study design, data collection and analysis, decision to publish, or preparation of the manuscript.

## Grant Disclosures

The following grant information was disclosed by the authors:
Medical Science and Technology Research Foundation of Guangdong: A2021335, PW.
Traditional Chinese Medicine Bureau of Guangdong Province: 20222166, PW.
Guangdong Provincial Basic, Applied Basic Regional Joint Fund: 2020A1515110009, PZ.
Guangzhou Science and Technology Project: 202102010060, ZL.
Guangzhou Science and Technology Bureau City School (Institute) Enterprise Joint Project: SL2024A03J00901.

## Competing Interests

The authors declare that they have no competing interests.

## Author Contributions

- Pengzhen Wang conceived and designed the experiments, performed the experiments, analyzed the data, prepared figures and/or tables, authored or reviewed drafts of the article, and approved the final draft.
- Pingping Zhu performed the experiments, analyzed the data, prepared figures and/or tables, authored or reviewed drafts of the article, and approved the final draft.
- Shaoheng Zhang conceived and designed the experiments, prepared figures and/or tables, authored or reviewed drafts of the article, and approved the final draft.
- Wei Yuan performed the experiments, prepared figures and/or tables, authored or reviewed drafts of the article, and approved the final draft.
- Zhihe Liu conceived and designed the experiments, prepared figures and/or tables, authored or reviewed drafts of the article, and approved the final draft.

## Data Availability

The raw data are available in the Supplemental Files.

## Supplemental Information

Supplemental information for this article can be found online at http://dx.doi.org/10.7717/peerj.15917#supplemental-information.

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
