# Peer review of "Icariin activates far upstream element binding protein 1 to regulate hypoxia-inducible factor-1α and hypoxia-inducible factor-2α signaling and benefits chondrocytes"

_PeerJ, doi:10.7717/peerj.15917_

## Round 0.1 · original submission · Major Revisions

On your submitted manuscript, the reviewers have provided useful comments. Please revise your paper according to these comments.

Reviewer 1 ·

Basic reporting

1) Overall, the findings of this study contribute to our understanding of the therapeutic effects and mechanisms of ICA in protecting chondrocytes and reducing inflammation, offering a promising avenue for the development of novel treatments for OA. However, there are some grammatical problems in the text, please go through and check the text. The manuscript needs a professional revision for the language.
2) The P value of * and # should be indicated in Fig 2 and Fig 4 legends. In addition, Figure legends of Fig 5 should be more detailed.
3) Is the“ACAN”in Figure 5 the same as the“AGG”elsewhere in the article? Please unify the expression in the full text. When the abbreviation first appears, the full names need to be provided.
4) The author should appropriately supplement references from the past 2 years.

Experimental design

5) The study found that ICA has a positive effect on chondrocytes, specifically in promoting ECM synthesis, proliferation and metabolism. Furthermore, the study highlighted FUBP1 as a key regulator in the response of chondrocytes to ICA treatment. Overexpression of FUBP1 enhanced the promoting effect of ICA, while its knockdown resulted in no significant difference in response to ICA treatment. Therefore, the study suggests targeting FUBP1 could be a potential therapeutic strategy for OA treatment utilizing ICA.
6) In Abstract, the author indicated that “The research aims at investigating the role and regulatory of far upstream element binding protein 1 (FUBP1) in chondrocytes treated by ICA to maintain a proper state and suppress inflammatory responses.” The author should illustrate why FUBP1 was selected for study in the manuscript.
7) The supplier information of materials used in this study should be provided. Please check for all the regents/drugs.

Validity of the findings

8) There is no need to repeat the description of the experimental method in the results section. Such as “Alcian blue staining is a classic methods for detecting the ECM component AGG [19]” and “Alcian blue and SO staining is a classic methods to detect the secretion of AGG (20).”
9) In Results, the author indicated that “chondrocyte viability was not affected by ICA at 2 μM (P>0.05)”. However, the result (Figure 1(c)) indicated that 0.5, 1 and 2 µM of ICA increased the AGG secretion of chondrocytes at varying degrees. The author should explain why 2 µM of ICA increased the AGG secretion of chondrocytes but 2 µM of ICA increased the AGG secretion of chondrocytes.

Additional comments

10)The discussion section of this study is very chaotic and requires a systematic and logical discussion based on the research purpose and findings. In addition, the limitations of this study need to be discussed in the discussion section.

Reviewer 2 ·

Basic reporting

a. This manuscript studied the effect and mechanism of ICA on chondrocyte extracellularmatrix (ECM) synthesis, proliferation and metabolism. In the study, The results revealed that ICA at the concentration of 1µM has the most significant effect on chondrocytes. The alcian blue staining and colony formation assays results showed the promotion effect of ICA was not significant in FUBP1 knockdown cells, and the promotion effect of ICA was significantly enhanced in FUBP1 overexpressed cells. ICA significantly increased the protein expression of FUBP1, HIF-1α, SOX9 and Col2α in chondrocytes with overexpressed FUBP1, and significantly inhibited the expression of HIF-2α and MMP9 in chondrocytes with knocked down FUBP1. It needs careful editing by someone with expertise in technical English editing paying particular attention to English grammar, spelling, and sentence structure so that the goals and results of the study are clear to the reader.

b. Overall, the structure of the article is relatively complete, the figures and tables basically meet the requirements, and the references are generally reasonable.

Experimental design

a. In Abstract, the author concluded that “In conclusion, ICA can be used as an effective drug for the treatment of OA with FUBP1 as a drug target.” The experiments and results in the article do not support this conclusion. Please modify these descriptions carefully.

b. In Introduction, the authors should Need A detailed description of research progress for the regulation of FUBP1 to HIF-1 α and HIF-2 α, and further demonstrating the significance of the role of FUBP1 in influencing chondrocytes by regulating HIF-1αand HIF-2 α.

c. In this study, chondrocytes were cultured in micromass culture and 3D scaffolds . What is the difference between the two methods.

Validity of the findings

a. In the“Cell and scaffolds 3D complex construction and culture”section, the complex blocks were harvested for 14 days. However, it is not clear whether the subsequent sections were paraffin or frozen sections, and the steps for subsequent staining with saffron and alcian blue are provided in detail.

b. The number of samples used in each experiment should be declared in the figure legends.

c. In Results, the author indicated that “Figure 2(a) showed that chondrocytes with FUBP1 (green) overexpression (FUBP1-OE) and knockdown (FUBP1-KD) were successfully constructed. The fluorescence intensity of FUBP1 (green) in FUBP1-OE and FUBP1-KD groups was stronger and weaker than that of the control groups, respectively.” The expression of FUBP1 needs to be validated by RT qPCR, WB, or IF to confirm whether FUBP1 is overexpressed or knockdown, and cannot be represented by fluorescence intensity.

Additional comments

During the discussion, the author should analyze and discuss the results by comparing them with other reports or papers. However, in the discussion section of this article, some of the descriptions are just repetitions of the result section, which is unnecessary and meaningless.

---

## Round 0.2 · Minor Revisions

Thank you for re-submitting the revised manuscript, however, I believe additional effort needs to go into further improving this article. Please revise the following issues:

1. Line 98: [In this study, we aimed to clarify, ….for future research] should be revised to [In this study, we aimed to clarify the relationship between FUBP1 and HIF-1α/2α in chondrocytes. This research will establish a solid foundation for future research].
2. Line 333: [slove] should be [solve].
3. Line 339: [in in Figure 5] should be [in Figure 5].
4. Line 339-341: [As summarized, …. respectively] should be revised to [As depicted in Figure 5, overexpression or silencing of FUBP1 regulates the HIF-1/2 signaling pathway to maintain chondrocyte activity or prevent chondrocyte inflammatory responses, respectively].

---

## Round 0.3 · accepted · Accept

By carefully evaluating the contents of this revised paper, I was satisfied with the responses and revisions made by the authors. With the necessary revisions and enhancements, the quality of this paper has been significantly improved. I believe this revised study is ready to be considered for publication in this Journal.